# Does Trans-Stimulation of L-Tyrosine Lead to an Increase in Boron Uptake in Head and Neck Squamous Cell Carcinoma Cells?

**Matthias Gielisch** , **Maximilian Moergel, Bilal Al-Nawas** and **Peer W. Kämmerer** *

Department of Oral- and Maxillofacial Surgery, University Medical Center Mainz, 55131 Mainz, Germany; matthias.gielisch@unimedizin-mainz.de (M.G.); maximilian.moergel@unimedizin-mainz.de (M.M.); bilal.al-nawas@unimedizin-mainz.de (B.A.-N.)
* Correspondence: peer.kaemmerer@unimedizin-mainz.de

**Abstract:** (1) Background: For advanced head and neck squamous cell carcinoma (HNSCC), boron neutron capture therapy (BNCT) is a potential therapeutic option, but high concentrations of boron within HNSCC are necessary. Therefore, this in vitro pilot study examined the uptake and concentration of boron in HNSCC cells using the trans-stimulation effect of L-tyrosine when compared to non-stimulated samples. (2) Methods: Two HNSCC cell lines were incubated with L-tyrosine for up to two hours, followed by incubation with three L-para-boronophenylalanine (BPA) concentrations (5, 20, 50 ppm) at eight incubation times (1–4.5 h in half-hour steps). Subsequently, cellular boron uptake was measured via inductively coupled plasma mass spectrometry. (3) Results: No differences between laryngeal and oral SCC cells were seen; therefore, data were pooled. In total, boron uptake was not significantly increased in trans-stimulated samples when compared to the control group (all $p > 0.05$). Nevertheless, with trans-stimulation, higher BPA concentrations resulted in higher intracellular boron concentrations (5 < 20 < 50 ppm; all: $p < 0.05$), whereas these differences were less distinct in the non-trans-stimulated group. (4) Conclusions: The effect of trans-stimulation for up to two hours seems to be less relevant for HNSCC, though trans-stimulated HNSCC cells seem to have a more distinct BPA-dose-dependent cellular boron uptake that might be addressed in further research.

**Keywords:** boron neutron capture therapy; trans-stimulation; head and neck cancer; squamous cell carcinoma; preloading

## 1. Introduction

Head and neck cancer—with about 90% squamous cell carcinoma (HNSCC)—accounts for 3% of all cancer types, with an incidence of about 900,000 new cases and 500,000 deaths per year [1]. Treatment, independent of stage, includes surgery, radiation and additional chemotherapy. However, for advanced as well as metastatic HNSCC, additional treatment options are needed. One of these therapies could be boron neutron capture therapy (BNCT), which is based on the reaction of the non-radioactive boron-10 with low-energy (thermal) neutrons, resulting in the nuclear reaction $^{10}B(n,\alpha)^7Li$. Because of the short range of the produced alpha particles and lithium from about 8–10 μm with a high linear energy transfer, the radiation effects mainly occur in cells containing boron [2]. To treat a tumour with BNCT, it is necessary to reach a high concentration of boron in tumour cells, especially in comparison to normal tissue. Second, the tumour should not be seated more than 6–8 cm under the skin, otherwise—due to its poor penetration—the thermal neutron beam cannot reach the tumour [3]. In practice, a boron carrier such as mercaptoundecahydro-closo-dodecaborate (BSH) or L-para-boronophenylalanine (BPA), a modified amino acid, is infused in a fructose solution into a peripheral vein, resulting in a higher uptake of the $^{10}B$ carrier in cancerous tissue as tumour cells have a higher turnover with an increased energy consumption [4,5]. The L-system transports amino acids with heavy side chains such as

tyrosine or phenylalanine [6]. Because of the similarity of BPA and phenylalanine, this transport system was identified as important for BPA uptake [7]. In the oral mucosa, LAT-1 is located in the basal cell layer, and is both intracellular and membrane-bound. Whereas the expression of LAT-1 in normal mucosa is low [8], it increases in cases of HNSCC and decreases for the highest dedifferentiation levels [9].

HNSCC is stated as a suitable target for BPA-mediated BNCT with a tumour/tongue $^{10}$B ratio of up to 4.1 [10] and a high rate of patients that respond to BNCT treatment, even with inoperable, locally recurrent and previously irradiated cancer [5]. Various approaches for enhancement of BPA uptake in carcinoma cells have been proposed, such as pre-treatments with low-dose gamma irradiation before or during BPA incubation [11] and pulsed focused ultrasound [12]. Different infusion methods for BPA such as intravenous, intra-arterial or direct infusion into the carotid artery in the case of brain tumours were studied as well [13]. Another way to increase boron uptake is based on the transport mechanism of amino acids that aims to reach an equilibrium between efflux and influx. In this regard, two relevant phenomena are described: cis-inhibition and trans-stimulation. Cis-inhibition, as seen in erythrocytes [14], inhibits productive signals, whereas trans-stimulation indicates an exchange of intracellular for extracellular amino acids via the L-system [15,16]. Trans-stimulation effects are described for various cell lines. For example, gliosarcoma cells showed an increased BPA accumulation after incubation with L-tyrosine, whereas simultaneous incubation of BPA and L-tyrosine led to a decrease in boron uptake [17]. In a xenograft tumour model, preloading with L-phenylalanine reduced the boron uptake in brain tissues relative to the uptake in the tumour tissue, and therefore reduced the maximum irradiation dose in normal tissue [18]. Further in vitro and in vivo investigations showed that preloading with L-tyrosine as well was with L-DOPA before exposure to BPA increased the intracellular boron concentration by a factor of three, while normal tissue did not show an increased BPA uptake [17,19,20].

The aim of the present study was an evaluation of the potential increase in the uptake of BPA into HNSCC cell lines utilising trans-stimulation with L-tyrosine, which seems to be clinically feasible. The null hypothesis was that trans-stimulation with L-tyrosine does not enhance the uptake of boron in HNSCC cells after BPA incubation.

## 2. Materials and Methods

### 2.1. Cell Culture

Two different head and neck squamous cell carcinoma (HNSCC) cell lines were used. Here, PCI 1 was cultured from a 65-year-old, male patient with a moderately differentiated, recurrent squamous cell carcinoma of the larynx with unknown TNM classification, whereas PCI 13 was cultured from a 50-year-old, male patient with a badly differentiated, recurrent squamous cell carcinoma of the trigonum retromolare with the following TNM classification: T3 N1 M0 [21]. Human oral keratinocytes (HOK, #2610, ScienCell, Carlsbad, CA, USA) and pooled human gingival epithelial cells (pHGEP; CELLnTEC, Bern, Switzerland) were used as a controls for immunoblotting.

Cell lines were cultured in Dulbecco's modified Eagle medium (DMEM; Sigma-Aldrich, St. Louis, MO, USA) supplemented with 10% foetal calf serum (GE Healthcare Life Sciences, Little Chalfont, UK), streptomycin and penicillin (GE Healthcare Life Sciences, Little Chalfont, UK) in culture flasks (250 mL, Greiner bioone, Kremsmünster, Austria) or cultures dishes (d = 100 mm, BD Biosciences, Franklin Lakes, NJ, USA). The incubator (Heraeus Instruments, Hanau, Deutschland) was set to humidified air containing 5% $CO_2$ at 37 °C. Cells were passaged in their log phase, approximately every 2–3 days.

### 2.2. Western Blot

Cells (PCI 1, PCI 13, HOK and pHGEP,) were cultured as described above in Petri dishes until approximately 80% confluence was reached. The following steps were conducted on ice: Cells were rinsed with cold phosphate-buffered saline (PBS) two times. Subsequently, cells were incubated with 500 μL radioimmunoprecipitation-buffer (RIPA

Buffer, R0278, Sigma-Aldrich, St. Louis, MO, USA), 50 μL protease-inhibition cocktail and 10 μL phosphatase-inhibition cocktail for 5 min. After mechanical release, protein concentrations were determined with a Pierce® BCA protein assay kit (REF: 23227, ThermoSCIENTIFIC, Waltham, MA, USA) according to the manufacturer's protocol. Finally, suspensions with 15 μg protein were used for separation in a Mini-PROTEAN® Tetra Cell (bio-rad Laboratories, Pleasanton, CA, USA) with a Mini-PROTEAN® TGX® Gel (#456-1094, bio-rad Laboratories). The samples were blotted on a PVDF membrane (IPVH00010, Immobilon®—P transfer membrane, Merck, Darmstadt, Germany). After blocking for 1 h, incubation with the primary antibody (#5347S, rabbit anti-LAT1 antibody, 1:1000, Cell Signaling Technology, Danvers, MA, USA) and Tris-buffered saline (TBS, Sigma-Aldrich) for 24 h took place. After washing three times with TBS-W (Sigma-Aldrich) for 15 min, the secondary HRP-linked antibody (#7074S, IgG anti-rabbit antibody, 1:1000, Cell Signaling Technology) was added for 1 h. Subsequently, the membrane was washed three times with TBS-W (Sigma-Aldrich) for 10 min. Enhanced luminol-based chemiluminescent (RPN2232, Amersham™, GE Healthcare Life Sciences, Little Chalfont, UK) was used for the detection of the horseradish peroxidase on the membrane. Analysis was conducted using the Western blot imaging system Fusion Solo S (Vilber Lourmat, Collégien, France).

### 2.3. Trans-Stimulation Experiments

As L-tyrosine and L-phenylalanine use the same (LAT-1) transport system, the decision to use L-tyrosine for these experiments was based on the results from other cell lines and trans-stimulation from other groups [19].

After 48 h of cultivation, medium was removed and cells were washed twice with phosphate-buffered saline (PBS) at 37 °C. Groups with one or two hours of trans-stimulation were incubated in a 5 mmol L-tyrosine solution. Here, a 50 mmol L-tyrosine stock-solution was prepared by stirring 180 mg L-tyrosine (Sigma-Aldrich, St. Louis, MO, USA) after filling it up to 20 g with Aqua Dest. After sterile filtration with a Whatman filter FP 30/0-2 CA-S (GE Healthcare Life Sciences, Little Chalfont, UK), the stock solution was diluted 10-fold with Hank's balanced salt solution (HBSS, Invitrogen Lifescience, Carlsbad, CA, USA). Incubation with L-tyrosine for trans-stimulation was conducted by removing the medium from cells after 48 h and rinsing two times with PBS at 37 °C. Afterwards, 2 mL of 5 mmol L-tyrosine solution was added to every dish and incubated for one or two hours in the incubator. Then, cells were covered with 2 mL of 5, 20 and 50 ppm BPA-solution (Katchem, Prague, Czech Republic) for 1, 1.5, 2, 2.5, 3, 3.5, 4 and 4.5 h (Figure 1). BPA solutions were produced by diluting a 500 ppm stock solution with HBSS.

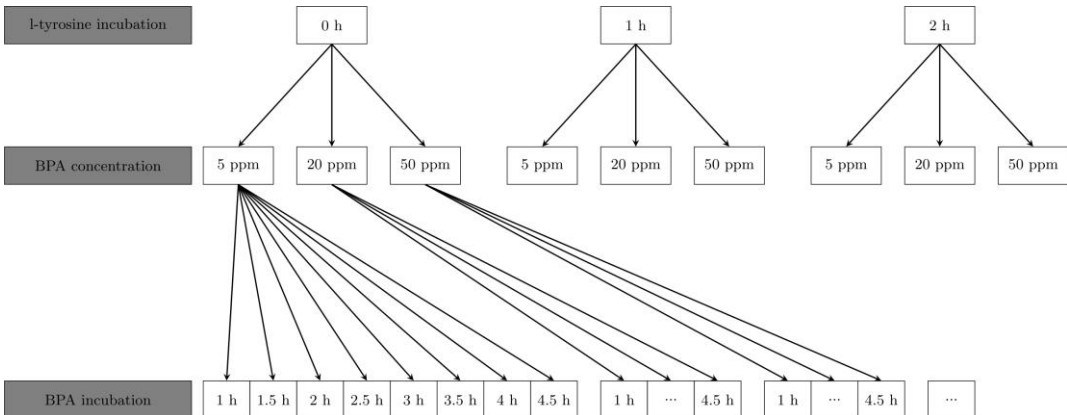

**Figure 1.** Flowchart of the different groups of L-tyrosine incubation times, used BPA concentrations and BPA incubation times; ppm = parts per million.

*2.4. Boron Analysis*

Acidic digestion for boron analysis was carried out using an inductively coupled plasma mass spectrometer (ICP-MS). The cell samples were prepared by adding 1 mL $HNO_3$ and 1 mL $H_2O_2$, followed by incubation on a vibrating plate. Afterwards, 1 mL of this solution was mixed with 1 mL $NH_3$, 2.9 mL of a solution containing 2% butanol, 1% $NH_3$, 0.05% EDTA and 0.05% Triton-X 100 (all from Sigma-Aldrich, Munich, Germany), 0.05 mL Rhodium Standard (1 ppm, High Purity Standards, North Charleston, SC, USA) and Beryllium Standard (10 ppm, High Purity Standards, North Charleston, SC, USA). Beryllium solution and rhodium solution were used as internal standards. For quantification, a calibration curve with BPA solutions were created. Finally, ICP-MS (Agilent 7500ce series (Santa Clara, CA, USA) was carried out in order to determine the intracellular boron content (for each group, at least n = 3).

*2.5. Statistics*

Data were collected with Microsoft Excel (Version 2013, Microsoft, Redmond, DC, USA) and statistics were carried out with RStudio (Version 1.2.1335, Integrated Development for R. RStudio, Inc., Boston, MA, USA) and R (Version 3.6.0, R Foundation for Statistical Computing, Vienna, Austria). Figures were created with R package ggplot2 (Version 3.2.1). Mean and standard deviations were calculated. Data from different cell lines, PCI 1 and PCI 13, and the two L-tyrosine incubation times, one and two hours, were analysed separately and in combination (pooled). Moreover, a multifactor analysis of variance (ANOVA) with a post hoc Games–Howell test was performed. Levene test for variance homogeneity was significant. Therefore, the following presented statistical results were considered to be descriptive. $p < 0.05$ was considered significant.

The control group was the non-trans-stimulated group, and the trans-stimulated group was defined as the treatment group. Independent variables were trans-stimulation with two factors, trans-stimulated versus non-trans-stimulated, BPA concentration with three factors, 5, 20 and 50 ppm, and BPA incubation time with eight factors. The dependent variable was the measured boron uptake.

### 3. Results

*3.1. Western Blotting of LAT-1 in PCI 1 and PCI 13 Cell Lines*

The expression of LAT-1 in PCI 1 and PCI 13 cell lines was confirmed by Western blotting (Figure 2). It can be seen that the amount of LAT-1 for healthy oral cells such as HOK and pHGEP is lower than for those of the tumour cell lines PCI 1 and PCI 13. Furthermore, the PCI 13 cell line showed a higher expression of LAT-1 when compared to PCI 1.

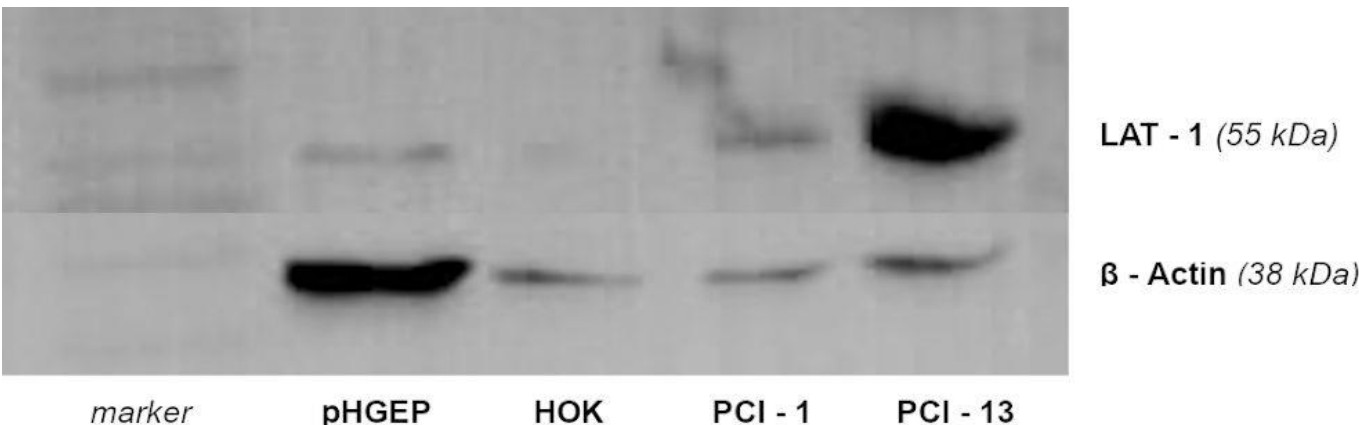

**Figure 2.** Results of Western blot for proteins from PCI 1 and PCI 13. LAT-1 is detectable for both cell lines at 38 kDa. Human oral keratinocytes (HOK) and pooled human gingival epithelial cells (pHGEP) were used as control. The expression is clearly higher in PCI 1 and PCI 13 when compared to controls.

*3.2. Boron Uptake in OSCC Cell Lines in Association with Trans-Stimulation*

As there were no statistically relevant differences when analysing the cell lines separately (all $p > 0.01$), the data of the two lines were pooled and described accordingly. The boron uptake depending on trans-stimulation and used BPA concentrations are shown in Figure 3.

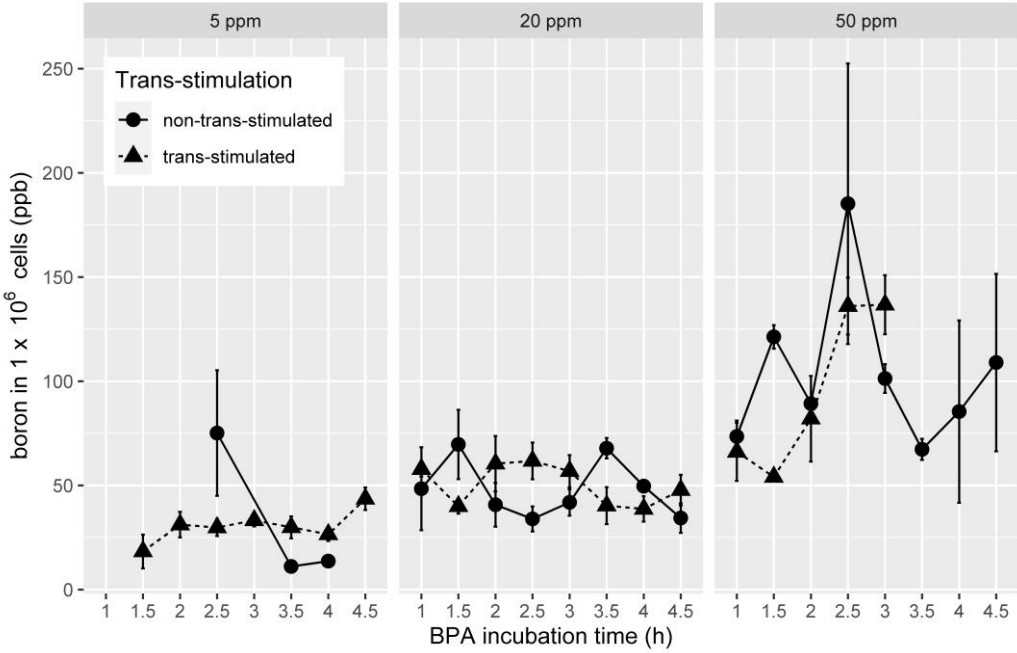

**Figure 3.** Line charts with error bars for standard error (SE) indicating boron content independent of BPA concentration (5, 20 and 50 ppm) and BPA incubation time (ppb = parts per billion, circles indicate non-trans-stimulated and triangles trans-stimulated cells).

In brief, no significant differences between both groups with the same BPA concentrations at the same time points were seen (all: $p > 0.05$, Table 1).

**Table 1.** Results of the ANOVA—no significant effect for trans-stimulation was seen.

| BPA Concentration (ppm) | Trans-Stimulation | Mean Boron Concentration (ppb per One Million Cells) | SD | *p* |
|---|---|---|---|---|
| 5 | no | 50.31600 | 63.72422 | 0.893 |
|   | yes | 29.25684 | 13.29931 |  |
| 20 | no | 48.92545 | 30.77410 | 0.998 |
|   | yes | 51.18492 | 26.31162 |  |
| 50 | no | 98.63308 | 49.07148 | 0.615 |
|   | yes | 82.42652 | 43.24766 |  |

For 5 ppm BPA concentration, the maximum boron content showed up after 2.5 h with 75.16 ppb per one million. For cells without trans-stimulation, the mean uptake for all time points was 50.32 (standard deviation (SD): ±63.72) ppb per one million cells. Trans-stimulated cells reached 43.60 ppb per one million cells after 4.5 h with a mean uptake of 29.27 (SD: ±13.30) ppb per one million cells for all time points. Incubation with a BPA concentration of 20 ppm resulted in the highest boron content (69.65 ppb per one million cells) in the non-trans-stimulated group after 1.5 h. With trans-stimulation, the highest boron content (61.78 ppb per one million cells) was seen after 2.5 h of BPA incubation. The mean uptake for all BPA incubation times with 20 ppm without trans-stimulation reached 48.93 (SD: ±30.77) ppb per one million cells and 51.18 (SD: ±26.31) ppb per one million cells with trans-stimulation. Regarding incubation with a BPA concentration of

50 ppm, the maximum was reached without trans-stimulation after 2.5 h (124.64 ppb per one million cells), whereas the mean uptake for all time points was 113.05 (SD: ±89.05) ppb per one million cells. In the trans-stimulated cells group, the highest boron uptake was detected after 3 h (136.74 ppb per one million cells) with a mean of 82.43 (SD: ±43.25) ppb per one million cells for all time points.

### 3.3. Boron Uptake in HNSCC Cell Lines in Association with BPA Concentration

With trans-stimulation, boron uptake between the various groups of BPA concentrations differed significantly, namely 20 vs. 50 ppm ($p < 0.032$, Table 2), 20 vs. 5 ppm ($p < 0.001$, Table 2) and 50 vs. 5 ppm ($p < 0.001$, Table 2). According to this, higher BPA concentrations resulted in higher intracellular boron concentrations. Without trans-stimulation, only the boron uptake in the group with a BPA incubation of 50 ppm was significantly higher in comparison with the boron uptake in the non-trans-stimulated 20 ppm group ($p < 0.012$, Table 2).

**Table 2.** Results of the Games–Howell post hoc test—significant differences were detected within the trans-stimulated groups.

| Trans-Stimulation | BPA Concentration (ppm) | Mean Boron Concentration (ppb per One Million Cells) | SD | *p* | | |
|---|---|---|---|---|---|---|
| | | | | 5 ppm vs. 20 ppm | 5 ppm vs. 50 ppm | 20 ppm vs. 50 ppm |
| no | 5 | 50.31600 | 63.72422 | 1.000 | 0.208 | 0.012 |
| | 20 | 29.25684 | 13.29931 | | | |
| | 50 | 48.92545 | 30.77410 | | | |
| yes | 5 | 51.18492 | 26.31162 | 0.000 | 0.000 | 0.032 |
| | 20 | 98.63308 | 49.07148 | | | |
| | 50 | 82.42652 | 43.24766 | | | |

## 4. Discussion

For BNCT, a high concentration of boron within the cancer cells is mandatory as, theoretically, only those cells will be destroyed after neuron irradiation, leaving surrounding normal cells undamaged. However, even if trans-stimulation seems to increase boron uptake in HNSCC cells with increasing BPA concentrations, no distinct effect of trans-stimulation for the uptake of boron in HNSCC cells after BPA incubation in comparison to non-stimulated cells could be shown.

For HNSCC, adjuvant radiation therapy and/or chemotherapy are recommended for advanced stages. Known adverse effects of radiation such as pain and xerostomia have a negative impact on speech, chewing and swallowing [22]. BNCT might reduce these side effects because of its principle to target tumour cells. Here, one major concern is that boron uptake in tumour cells has to be increased, as reported disadvantages of BNCT include inhomogeneous BPA distribution in the tumour and an insufficient coverage of the clinical target volume [23]. The expression of LAT-1, which is proposed to be one of the most relevant amino acid transporters for BPA uptake, increases from normal to precursor lesions to tumour tissue, but seems to decrease for higher dedifferentiation levels of HNSCC [9]. Therefore, the HNSCC cell lines used in the present study are described as moderately and badly differentiated [21] and the presence of LAT-1 in both HNSCC cell lines was confirmed by Western blotting.

The BPA uptake of HNSCC cells in this analysis was 1.10 times higher after trans-stimulation with a BPA concentration of 50 ppm when compared to the non-trans-stimulated samples, but without a significant difference. In the literature, the boron content of B16F1 melanoma cells was increased after trans-stimulation to 1.8–2.7 times higher [19] and in GS-9L gliosarcoma cells to approximately two times higher [17]. A reason for the non-existent effect of trans-stimulation in HNSCC cells could be the dedifferentiation of the tumour cells, which leads to a lower LAT-1 expression [9]. Overall, beneath the trans-stimulation effect, it is also possible that the lack of leucin, an amino acid similar to BPA, leads to an activation of system L [24,25]. Subsequently, the observed minor increase in boron content

could be justified by this activation, because the trans-stimulation medium was free of amino acids except L-tyrosine.

It is also remarkable that the overexpression of system L transporter LAT-1 in different cells such as hepatocytes and fibroblasts did lead to increased leucin uptake in hepatocytes but not in fibroblasts [26]. It appears likely that miscellaneous cell lines with their proteomes show different uptake behaviour. For example, in human hepatocellular carcinoma cells, no trans-stimulation effect was seen [27]. Likewise, melanoma cells showed no significantly increased boron content after trans-stimulation, and a lower boron uptake after 90 min of trans-stimulation was observed for adenocarcinoma cells and human hepatocellular carcinoma cells [28]. Most commonly, the concentration of the used L-tryrosine solution was 5 mmol/L, as this concentration is known to be non-toxic in vitro. Therefore, this concentration was used in the present analysis as well. However, a solution with 2.5 mmol seems to improve the boron uptake after trans-stimulation in certain cell lines [28].

Research using longer trans-stimulation times with L-DOPA for rat glioma cells achieved the strongest trans-stimulation effect after 4 h, whereas a small increase in the resulting BPA concentration was seen after two hours of stimulation [20]. Another investigation with shorter trans-stimulation times up to 90 min indicated a decrease in the measured boron uptake after 1 h for some cell lines, whereas for others the uptake did not change between 60 and 90 min [28]. Since this seemed to be a critical time period, the effect of up to two hours of incubation time with L-tyrosine was investigated in the present study without detecting significant differences. However, longer incubation times of up to 4 h, as demonstrated by Yang et al. for glioblastoma and melanoma cells [13], might have resulted in higher BPA concentrations, even if this effect was not seen by the same group in vivo.

The timepoint of the highest boron increase without trans-stimulation was seen in the time between 1.5 and 2.5 h and is analogous to the results described in the literature [10,29]. A significantly higher boron uptake without trans-stimulation was reached by administration of high BPA concentrations (20 vs. 50 ppm, $p < 0.012$). Healthy tissues such as skin also show a higher boron uptake depending on the BPA concentration. In lung cancer metastasis [30] and HNSCC in a hamster cheek pouch model [29], a higher BPA concentration resulted in higher boron uptake. Interestingly, the use of L-tyrosine led to significant differences between the three BPA concentration groups (5 vs. 20 vs. 50 ppm). The effect between the 5 and 20 ppm groups seems to arise from a lower boron uptake in the 5 ppm group. Besides the effect of trans-stimulation, another effect, called trans-inhibition, was described as a negative feedback regulation [31]. It is supposed that the mechanism behind this effect relies on a slowing of entry by a high intracellular content, i.e., of an analogue of the solute (in this case L-tyrosine). The observed lower boron uptake could be the result of this effect. Hence, the use of L-tyrosine in combination with low BPA concentrations could be counterproductive for an increase in boron uptake. Future research might need to focus on boron uptake when comparing more than one cell line that is moderately differentiated with more than one cell line that is weakly differentiated.

## 5. Conclusions

The combination of trans-stimulation and an increased BPA dose seem to have a relevant effect on boron uptake by HNSCC cells. As a consequence, the efficiency of BNCT to treat recurrent or enlarged NHSCC could be enhanced via selective enrichment of boron in tumour cells, even if the tumour and the normal cells are mingled at the tumour margin. Even so, the effect of trans-stimulation for up to two hours seems to be less relevant for moderately to badly differentiated HNSCC, and might not be a sufficient method for boron enrichment in those tumour entities.



**Author Contributions:** Conceptualisation, M.G., B.A.-N., M.M. and P.W.K.; methodology, M.G., B.A.-N. and M.M.; validation, M.G., B.A.-N. and M.M.; formal analysis, M.G., B.A.-N., P.W.K. and M.M.; investigation, M.G.; resources, B.A.-N.; data curation, M.G.; writing—original draft preparation, M.G. and P.W.K.; writing—review and editing, M.G. and P.W.K.; visualisation, M.G. and P.W.K.; supervision, M.M. and B.A.-N.; project administration, M.M. and B.A.-N.; funding acquisition, M.G. All authors have read and agreed to the published version of the manuscript.

**Funding:** This research received no external funding.

**Institutional Review Board Statement:** Not applicable.

**Informed Consent Statement:** Not applicable.

**Data Availability Statement:** The data presented in this study are available on request from the corresponding author.

**Conflicts of Interest:** The authors declare no conflict of interest.

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
