# Peer review of "Does Trans-Stimulation of L-Tyrosine Lead to an Increase in Boron Uptake in Head and Neck Squamous Cell Carcinoma Cells?"

_applsci, doi:10.3390/app11167286_

Round 1

Reviewer 1 Report

The manuscript titled “Does trans-stimulation of l-tyrosine lead to an increase of boron uptake in head and neck squamous cell carcinoma cells?” investigated the uptake of boron in the form of BPA in HNSCC following trans-stimulation with 1-tyrosine. The findings in this study did not showed a robust uptake of boron with trans-stimulation.

Suggested revisions:

  • I do not fully understand the rationale for combining the two HNSCC cell lines to determine boron uptake following trans-stimulation with 1-tyrosine. Especially since the two cell lines have significantly different levels of LAT-1 which the authors indicate is necessary for boron uptake. The authors may have seen a better representation of boron uptake by separating the two cell lines, as I would expect the weakly differentiated cell line, PCI-13, to show a robust increase in boron uptake compare the moderately differentiated cell line, PCI-1. Additionally, it would have been nice for the authors to use two cell lines that are moderately differentiated (expressing similar levels of LAT-1) and two cell lines that are weakly differentiated (expression similar levels of LAT-1) to do their trans-stimulation study.
  • Please state the rationale for not using phenylalanine since the transport system is important for BPA uptake.

Author Response

These are the detailed comments on the reviewers’ suggestions and remarks. Revised text within the manuscript is marked in RED.

First of all, we would like to thank the reviewers for their effort and their time in order to augment our manuscript. We do know that research without groundbreaking results might be difficult to be published but we think that this should also be implemented. In brief, we really appreciate your valuable effort!

Reviewer 1:

Comment #1: “I do not fully understand the rationale for combining the two HNSCC cell lines to determine boron uptake following trans-stimulation with l-tyrosine. Especially since the two cell lines have significantly different levels of LAT-1 which the authors indicate is necessary for boron uptake. The authors may have seen a better representation of boron uptake by separating the two cell lines, as I would expect the weakly differentiated cell line, PCI-13, to show a robust increase in boron uptake compare the moderately differentiated cell line, PCI-1.

Our response: We totally agree. At first, we separated the two cell lines (which are also from two different entities) and analyzed the data in accordance. Though, the results did not show a difference in boron uptake. Therefore, in order to obtain higher numbers and make statistical comparisons more feasible, we pooled the data. This is explained in the manuscript now.

Comment #2: Additionally, it would have been nice for the authors to use two cell lines that are moderately differentiated (expressing similar levels of LAT- 1) and two cell lines that are weakly differentiated (expression similar levels of LAT-1) to do their transstimulation study.

Our response: We do agree to this important remark. Unfortunately, the analysis systems are not available anymore. Therefore, we could not carry out additional experiments. Though, this relevant bias was added to the discussion section.

Comment #3: “Please state the rationale for not using phenylalanine since the transport system is important for BPA uptake.”

Our response: As l-tyrosine and l-phenylalanine are using the same (LAT-1) transport system, our decision to use l-tyrosine is based on the results with other cell lines (melanoma cells) and trans-stimulation from other groups (mostly Papaspyrou et al., 1994), who have seen an increased BPA-uptake with l-tyrosine. This was added to the Materials & Methods section.

Reviewer 2 Report

The effect of trans-stimulation of OSCC cells as well as the role of para-boronophenylalanine (BPA) on boron neutron capture therapy (BNCT) was analyzed. This strategy is an additional therapeutic option in recurrent OSCC. In the introduction, some more details to the BNCT are necessary. In addition, the fabulous review by Suzuki (Suzuki M, Int J Clin Oncol 2020 Jan;25(1):43-50. , PMID 31168726) should be cited.

The conclusion is very short and translational aspects are missed. For instance: Both, the trans-stimulation as well as increased BPA doses have an effect on boron uptake by OSCC-cells. As a consequence, the efficiency of BNCT to treat recurrent or enlarged OSCC´s can be enhanced due to the selective enrichment of boron in tumor cells even if tumor and normal cells are mingled at the tumor margin.

In general, a list of abbreviations would further enhance the manuscript. Reference list: line 334 – “1.” should be omitted.

Author Response

These are the detailed comments on the reviewers’ suggestions and remarks. Revised text within the manuscript is marked in RED.

First of all, we would like to thank the reviewers for their effort and their time in order to augment our manuscript. We do know that research without groundbreaking results might be difficult to be published but we think that this should also be implemented. In brief, we really appreciate your valuable effort!

Reviewer 2:

Comment #1: “In the introduction, some more details to the BNCT are necessary. In addition, the fabulous review by Suzuki (Suzuki M, Int J Clin Oncol 2020 Jan;25(1):43-50, PMID 31168726) should be cited.“

Our response: We tried our best to add some more details and we cited the very well-written and informative article.

Comment #2: “The conclusion is very short and translational aspects are missed. For instance: Both, the trans-stimulation as well as increased BPA doses have an effect on boron uptake by OSCC-cells. As a consequence, the efficiency of BNCT to treat recurrent or enlarged OSCC´s can be enhanced due to the selective enrichment of boron in tumor cells even if tumor and normal cells are mingled at the tumor margin.“

Our response: Thank you very much for your suggestion. We hope that you do not mind that we implemented this into the manuscript.

Comment #3: “In general, a list of abbreviations would further enhance the manuscript.

Our response: We tried to avoid non-relevant abbreviations.

Comment #4: „Reference list: line 334 – “1.” should be omitted.“

Our response: Done.

Round 2

Reviewer 1 Report

The manuscript titled “Does trans-stimulation of l-tyrosine lead to an increase of boron uptake in head and neck squamous cell carcinoma cells?” investigated the uptake of boron in the form of BPA in HNSCC following trans-stimulation with 1-tyrosine. The edits improve the quality of the manuscript from its original form. The findings in this study did not showed a robust uptake of boron with trans-stimulation.

Suggested revisions:

  • Please add a space between “effect” and “on” in line 252 of the manuscript.

Author Response

Suggested revisions:

  • Please add a space between “effect” and “on” in line 252 of the manuscript.

Our comment: changes were done, thank you!
